# Investigation of Anticonvulsant Potential of *Morus alba*, *Angelica archangelica*, *Valeriana officinalis*, and *Passiflora incarnata* Extracts: In Vivo and In Silico Studies

**DOI:** 10.3390/ijms26136426

**Published:** 2025-07-03

**Authors:** Felicia Suciu, Dragos Paul Mihai, Anca Ungurianu, Corina Andrei, Ciprian Pușcașu, Carmen Lidia Chițescu, Robert Viorel Ancuceanu, Cerasela Elena Gird, Emil Stefanescu, Nicoleta Mirela Blebea, Violeta Popovici, Adrian Cosmin Rosca, Cristina Isabel Viorica Ghiță, Simona Negres

**Affiliations:** 1Faculty of Pharmacy, “Ovidius” University of Constanta, 900470 Constanta, Romania or felicia.suciu@drd.umfcd.ro (F.S.); nicoleta.blebea@365.univ-ovidius.ro (N.M.B.); cosmin.rosca@univ-ovidius.ro (A.C.R.); 2Faculty of Pharmacy, “Carol Davila” University of Medicine and Pharmacy, 020956 Bucharest, Romania; anca.ungurianu@umfcd.ro (A.U.); corina.andrei@umfcd.ro (C.A.); ciprian.puscasu@umfcd.ro (C.P.); robert.ancuceanu@umfcd.ro (R.V.A.); cerasela.gird@umfcd.ro (C.E.G.); emil.stefanescu@umfcd.ro (E.S.); simona.negres@umfcd.ro (S.N.); 3Faculty of Medicine and Pharmacy, Dunărea de Jos University of Galaţi, 800010 Galați, Romania; carmen.chitescu@ugal.ro; 4Center for Mountain Economics, “Costin C. Kritescu” National Institute of Economic Research (INCE-CEMONT), Romanian Academy, 725700 Vatra-Dornei, Romania; 5Faculty of Medicine, “Carol Davila” University of Medicine and Pharmacy, 050474 Bucharest, Romania; isabel.ghita@umfcd.ro

**Keywords:** medicinal plant extracts, bioactive phytocompounds, epilepsy, antiseizure, motor activity, molecular docking, 5-HT3 receptor, GABA-A receptor, integrin α4β1

## Abstract

The current study evaluated the anticonvulsant properties of ethanolic extracts from *Morus alba, Angelica archangelica*, *Passiflora incarnata*, and *Valeriana officinalis* using integrated phytochemical, in vivo, biochemical, and computational approaches. Phytochemical analysis by UHPLC-HRMS/MS revealed the presence of various bioactive compounds, notably flavonoids such as isorhamnetin, quercetin, and kaempferol. In an electroshock-induced seizure model, *Morus alba* extract (MAE, 100 mg/kg) demonstrated significant anticonvulsant effects, reducing both seizure duration and incidence, likely mediated by flavonoid interactions with GABA-A and 5-HT3A receptors, as suggested by target prediction and molecular docking analyses. The extracts of *Angelica archangelica* (AAE, 100 mg/kg) and *Passiflora incarnata* (PIE, 50 mg/kg) exhibited moderate, non-significant anticonvulsant activities. At the same time, *Valeriana officinalis* (VOE, 50 mg/kg) displayed considerable antioxidant and anti-inflammatory properties but limited seizure protection. All extracts significantly reduced brain inflammation markers (TNF-α) and enhanced antioxidant defenses, as indicated by total thiols. Molecular docking further supported the interaction of key phytochemicals, including naringenin and chlorogenic acid, with human and mouse 5-HT3A receptors. Overall, *Morus alba* extract exhibited promising therapeutic potential for epilepsy management, warranting further investigation into chronic seizure models and optimized dosing strategies.

## 1. Introduction

Epilepsy is a chronic neurological disorder characterized by recurrent, unprovoked seizures, affecting approximately 50 million individuals worldwide and presenting significant cognitive, psychological, and social burdens [1,2]. The global prevalence and complex nature of epilepsy highlight substantial challenges in disease management, underscored by the limitations of current therapeutic approaches. Despite extensive pharmacological advancements, around 30% of patients experience pharmacoresistance, and many others endure adverse effects from antiepileptic drugs (AEDs), such as sedation, cognitive impairment, hepatotoxicity, and behavioral changes, compromising long-term adherence [3,4,5]. Therefore, there is an urgent and continuous demand for novel therapeutic interventions characterized by higher tolerability and fewer adverse reactions.

In recent years, medicinal plants and their bioactive compounds have garnered significant attention as sources of new therapeutics due to their accessibility, safety, and efficacy [6]. Researchers are increasingly exploring plant-derived substances for a range of biomedical applications, from novel antimicrobials to anticancer agents, as these natural products often demonstrate potent effects with fewer side effects [6,7]. This surge of recent studies underlines the modern renaissance of phytotherapy in scientific research, situating natural compounds at the forefront of drug discovery. For instance, flavonoids have been spotlighted in the recent literature as promising plant-based therapeutics. Prunin, a flavanone glycoside found in citrus fruits and plants of the *Prunus* genus, was recently highlighted as an emerging anticancer compound [8]. Rana and Mumtaz (2025) [8] reported that prunin can promote apoptosis, inhibit tumor cell growth, and prevent tumor development by modulating key signaling pathways, underscoring its potential as a plant-derived anticancer agent. Notably, the biomedical significance of prunin has only come to light in the last couple of years, and its molecular mechanisms are just beginning to be unraveled [8]. This example illustrates how even well-known phytochemicals are now being explored in depth to uncover their therapeutic properties, highlighting that many plant-based molecules remain under-investigated despite their promising bioactivities.

Medicinal plants represent a compelling avenue for adjunctive epilepsy therapy, attributed to their complex mixtures of bioactive constituents that can target multiple pathological pathways implicated in seizure generation and propagation [5,9,10,11]. Herbal treatments have historically been integrated into various traditional medicinal systems. They are currently gaining attention in scientific research for their potential as anticonvulsants, as well as their favorable safety profiles compared to synthetic AEDs [12,13].

*Morus alba* L. (white mulberry), widely utilized in traditional Chinese medicine, has demonstrated promising anticonvulsant and neuroprotective activities [14]. Its therapeutic effects are primarily attributed to flavonoids, such as quercetin, kaempferol, and isorhamnetin, which exhibit antioxidant properties and modulate GABAergic neurotransmission [15,16]. A recent study reported that *M. alba* leaf infusions from natural sources contain remarkably high levels of chlorogenic acid and flavonoids, correlating with strong antioxidant activities determined using in vitro and ex vivo models [17]. In preclinical studies, *M. alba* extracts have been shown to delay seizure onset and elevate seizure thresholds, potentially reducing seizure severity and neuronal damage through combined antioxidant and inhibitory neurotransmission mechanisms [18,19].

*Angelica archangelica* L. (garden angelica), traditionally employed for neurological conditions, is enriched with coumarins, essential oils, and furanocoumarins [20,21]. These constituents confer potent anti-inflammatory, neuroprotective, and anticonvulsant properties. Essential oil from *A. archangelica* has been shown to reduce seizure severity and mortality rates in animal models of pentylenetetrazole (PTZ)-induced seizures, potentially through modulation of calcium channel activities and nitric oxide synthesis pathways, thereby exhibiting therapeutic relevance for epilepsy [22,23].

*Passiflora incarnata* L. (passionflower) is renowned for its anxiolytic and anticonvulsant efficacy. The plant’s primary active constituents, including flavonoids such as vitexin and isovitexin, modulate central GABAergic systems, thereby exerting anxiolytic, sedative, and anticonvulsant activities [24,25]. Experimental evidence has shown that *P. incarnata* extracts significantly prolong seizure latency and reduce seizure frequency in PTZ- and maximal electroshock (MES)-induced seizure models, primarily through modulation of the GABA-A receptor and antioxidant effects [26,27].

*Valeriana officinalis* L. (valerian) is traditionally used to treat insomnia and anxiety disorders. It contains valerenic acid, sesquiterpenes, and iridoid glycosides, which potentiate GABAergic neurotransmission, providing anticonvulsant and sedative effects [28,29,30]. Valerian extracts have been shown to effectively suppress seizures in MES and PTZ-induced animal models, suggesting significant potential as a complementary anticonvulsant therapy, particularly in drug-resistant epilepsy [31,32,33].

The selection of *Morus alba*, *Angelica archangelica*, *Passiflora incarnata*, and *Valeriana officinalis* for this study was guided by their phytochemical profiles, particularly their content in bioactive compounds capable of modulating γ-aminobutyric acid (GABA) signaling. GABA is the primary inhibitory neurotransmitter in the central nervous system, playing a crucial role in modulating pain, anxiety, and seizure thresholds through its actions on GABA-A and GABA-B receptors [34]. GABA-A receptors mediate fast synaptic inhibition via ligand-gated chloride channels, while GABA-B receptors modulate slow inhibitory transmission through G-protein-coupled mechanisms. Both receptor subtypes have been implicated in the pharmacological control of seizures and analgesia. *Morus* species have been reported to contain endogenous GABA in their leaves, with *M. alba* L. showing a concentration of 0.64%, comparable to other species such as *M. atropurpurea* and *M. mongolica* [35]. Moreover, morusin, a prenylated flavonoid isolated from *M. alba*, has demonstrated anticonvulsant activity via modulation of GABA receptors [19]. *Passiflora incarnata* extracts have also been shown to elicit robust GABAergic currents in hippocampal neurons, attributed in part to their naturally high GABA content, and to exert anxiolytic and anticonvulsant effects in vivo [25]. *Angelica archangelica* contains several coumarin derivatives (e.g., imperatorin, osthol, and cnidilin) that have been shown to potentiate GABA-induced chloride influx in GABA-A receptor-expressing systems in a concentration-dependent manner [36]. Lastly, *Valeriana officinalis* and its main active constituent, valerenic acid, have been demonstrated to modulate GABA-A receptor function and enhance the activity of central nervous system depressants and anesthetics [37]. Collectively, the pharmacological profiles of these four species provide a strong rationale for their investigation in the context of anticonvulsant therapy.

The current study aimed to investigate the anticonvulsant potential of four ethanolic extracts of *M. alba, A. archangelica, V. officinalis*, and *P. incarnata* through an integrated phytochemical profiling approach, in vivo evaluation using an electroshock-induced seizure model, biochemical assays measuring inflammatory and oxidative stress markers, and in silico analyses. Therefore, this paper sought to provide a comprehensive assessment of the therapeutic potential in epilepsy management, with a particular focus on identifying potential mechanisms underlying the neuroprotective effects of these medicinal plants.

## 2. Results

### 2.1. Phytochemical Characterization

A total of 26 phytochemicals were identified in the tested extracts based on exact mass and adduction matching using high-resolution UHPLC-HRMS/MS analysis. These included flavonoids (e.g., quercetin, kaempferol, apigenin, isorhamnetin), phenolic acids (e.g., chlorogenic acid, ferulic acid, gallic acid), and isoflavones (e.g., daidzein, genistein, formononetin). Compound identification was confirmed by comparing the exact masses of deprotonated molecular ions and corresponding adducts with theoretical values from spectral libraries, with a mass error of less than 5 ppm (Table 1).

The UHPLC-HRMS/MS chromatographic profiles revealed distinct sets of phytochemicals across the four analyzed plant extracts (Appendix A Appendix A). In the *Angelica archangelica* extract (AAE), the identified compounds included gallocatechin, quinic acid, feruloylquinic acid, aesculetin, daphnin, isorhamnetin-3-O-hexoside, and azelaic acid. The *Morus alba* extract (MAE) showed a diverse set of compounds, including catechin, daidzein, baptigenin, 2′-hydroxygenistein, and a group of structurally related flavonoids: alopecurone A, lechianoine A, prostatrol F, and sifroronol A. The *Passiflora incarnata* extract (PIE) was rich in neochlorogenic acid, chlorogenic acid, biochanin A glucoside, diosmetin 7-O-rutinoside, kaempferol-O-glucoside, procyanidin, caffeoylshinic acid, and baptigenin. The phytochemical profile of *Valeriana officinalis* extract (VOE) included caffeoylshinic acid, apigenin 7-O-glucosylglucoside, oleanolic acid, hispidulin 7-rutinoside, hispidulin, rutin, and valerenic acid. These findings highlight the chemical diversity among the studied plant species and support their relevance for pharmacological evaluation.

The concentration of major compounds varied substantially among the four extracts (Table 2). MAE displayed the highest phytochemical complexity and content, with particularly high concentrations of isorhamnetin (61.936 mg/g, 6.2% of exact mass), quercetin (26.357 mg/g, 2.6% of exact mass), kaempferol (7.936 mg/g), and catechin (10.736 mg/g). Other notable compounds in MAE included chlorogenic acid (1.761 mg/g), hyperoside (2.368 mg/g), and p-coumaric acid (2.091 mg/g). PIE was rich in flavonoids, including quercetin (5.661 mg/g), isorhamnetin (1.885 mg/g), and p-coumaric acid (3.155 mg/g). It also contained moderate levels of rutin (289.39 µg/g), genistin (93.45 µg/g), and apigenin (40.75 µg/g). AAE showed high levels of ferulic acid (6.351 mg/g, 0.64% of extract), chlorogenic acid (779.76 µg/g), and p-coumaric acid (1.223 mg/g). In comparison, apigenin (7.68 µg/g) and quercetin derivatives were present in lower amounts. VOE contained fewer compounds overall, with chlorogenic acid (11.714 mg/g, 1.17% of exact mass) and p-coumaric acid (1.099 mg/g) being the most abundant. Rutin (123.53 µg/g) and gallic acid (207.59 µg/g) were also detected at moderate concentrations. Some compounds, such as epicatechin gallate, glycitein, and hesperetin, were not detected in any of the extracts under the used conditions.

### 2.2. In Vivo Studies

#### 2.2.1. Effects on Motor Activity

Initial assessments of horizontal and vertical locomotor activity showed no statistically significant differences among groups, confirming baseline homogeneity (one-way ANOVA, *p* > 0.05). All mice displayed comparable spontaneous activity before treatment (Figure 1A,B).

After 6 days of treatment, no significant differences in motor activity were observed between treatment groups for either horizontal or vertical movement (one-way ANOVA, *p* > 0.05). Slight, non-significant increases in horizontal activity were noted for the AAE (+11.97%) and PIE (+10.61%) groups, while MAE (−13.53%) and VOE (−12.25%) induced modest reductions (Figure 1C). Phenobarbital led to a minor increase in horizontal movement (+4.25%). However, vertical activity showed more variability (Figure 1C). MAE reduced movement by 43.81%, while VOE and AAE produced small increases (+12.89% and +9.79%). PIE and phenobarbital showed minor decreases of −2.06% and −6.70%, respectively. None of these changes reached statistical significance (*p* > 0.05).

Thirty minutes after seizure induction, horizontal locomotor activity significantly increased in the phenobarbital group compared to controls (+91.99%; Kruskal–Wallis *p* = 0.0135, Dunn’s post hoc *p* = 0.0210, Figure 1D). No significant differences were observed for AAE (−13.74%, *p* > 0.05), MAE (−14.25%, *p* > 0.05), PIE (+6.66%, *p* > 0.05), or VOE (+10.62%, *p* > 0.05). These results indicate that seizure induction could have led to a decrease in spontaneous motor activity for all treatment groups except for PHB, suggesting that phenobarbital prevented these changes. Vertical activity remained statistically unchanged across all groups (one-way ANOVA, *p* > 0.05, Figure 1E). AAE caused the most significant reduction (−81.62%), followed by MAE (−36.94%) and PIE (−20.72%). VOE and phenobarbital led to slight, non-significant increases in activity (+8.88% and +13.51%).

#### 2.2.2. Anti-Seizure Activity

The anticonvulsant effects of the treatments were assessed after 7 days of administration using an electroshock-induced seizure model (15 mA, 3 s). Phenobarbital (25 mg/kg) completely abolished tonic-clonic seizures, reducing seizure duration by 100% compared to control (Kruskal–Wallis *p* = 0.0006, Dunn’s post-hoc *p* = 0.0018, Figure 2A).

Among the tested plant extracts, MAE demonstrated a strong anticonvulsant effect, reducing mean seizure duration by 92.86% (*p* = 0.0242 vs. control, Dunn’s post-hoc test). AAE and PIE produced moderate, non-significant reductions in seizure duration (−44.96% and −54.89%, respectively; *p* > 0.05). VOE had minimal impact (−3.33%; *p* > 0.05).

Seizure incidence analysis revealed a 100% convulsion rate in control animals, whereas phenobarbital provided complete protection (0% incidence; Fisher’s exact test, *p* < 0.0001). MAE significantly reduced seizure incidence to 50% (*p* = 0.0385) compared to the control. The AAE and VOE groups had convulsion rates of 87.5%, while the PIE group reduced the incidence to 62.5%. However, none of these differences reached statistical significance (*p* > 0.05) (Figure 2B).

Lethality rates post-seizure were 25% in the control group and 0% in the phenobarbital group. However, this reduction was not statistically significant (*p* > 0.05, Fisher’s exact test). Administration of MAE and PIE resulted in lethality rates similar to those of the control (25%). AAE produced a slightly higher rate (37.5%), and VOE showed a lower rate (12.5%). However, none of these differences were statistically significant (Figure 2C).

Overall, MAE demonstrated the most notable anticonvulsant activity by significantly reducing both seizure duration and incidence. It aligns with previous reports that attribute the effect to the modulation of GABAergic neurotransmission by *Morus alba* constituents [12,13]. Although AAE and PIE showed partial efficacy, the doses used in this study may have been insufficient to achieve statistically significant effects.

#### 2.2.3. Biochemical Assays

Regarding TNF-α levels in brain tissue, ANOVA analysis revealed significant differences among the groups (*p* < 0.0001; Figure 3A). PHB group and all groups treated with natural extracts showed a substantial reduction in TNF-α levels compared to the control group (*p* < 0.001, Bonferroni post-hoc test). The groups receiving plant extracts demonstrated reductions comparable to the PHB group, with the most pronounced decrease observed in the VOE group (39.64%), followed by the AAE group (36.34%), PIE group (34.93%), PHB group (32.91%), and MAE group (30.69%), relative to the CTL group.

After 7 days of treatment, the total thiol concentration was assessed in brain tissue samples. Biochemical analysis revealed significant differences among the experimental groups (univariate ANOVA, *p* < 0.0001; Figure 3B). No significant difference was observed between CTL and PHB. In contrast, all four groups treated with natural extracts showed increased levels of total thiols, expressed as glutathione (GSH) equivalents, when compared to the CTL group. However, the increase reached statistical significance only in the PIE and VOE groups, with rises of 102.05% and 93.12%, respectively.

### 2.3. Computational Studies

#### 2.3.1. Target Prediction

To elucidate the potential mechanisms underlying the observed anticonvulsant effects of the plant extracts, we employed the Super-PRED tool to predict molecular targets of the identified phytochemicals, with a focus on those associated with epilepsy. The analysis revealed several compounds with high-probability interactions with targets involved in seizure modulation.

Multiple flavonoids demonstrated predicted interactions with the GABA-A receptor (Table 3), specifically the α1β2γ2 subunit configuration: pinostrobin (0.7796), formononetin (0.7516), galangin (0.7201), chrysin (0.7022), epicatechin gallate (0.6461), glycitein (0.6125), and daidzein (0.5649). The GABA-A receptor is a pivotal inhibitory neurotransmitter receptor in the central nervous system. Positive allosteric modulation of this receptor enhances inhibitory neurotransmission, thereby reducing neuronal excitability and susceptibility to seizures [38]. Clinically, several antiepileptic drugs, including benzodiazepines and barbiturates, exert their effects through this mechanism. The predicted interactions suggest that these flavonoids may contribute to the anticonvulsant properties observed in the plant extracts by modulating GABAergic neurotransmission.

Two compounds, epicatechin gallate (probability: 0.6823) and pinocembrin (0.5441), were predicted to interact with the 5-HT3 receptor, a ligand-gated ion channel subtype of serotonin receptors. Activation of 5-HT3 receptors has been shown to exert anticonvulsant effects in various models of epilepsy, potentially by modulating neurotransmitter release and neuronal excitability [39,40]. These interactions may further contribute to the seizure-modulating effects of the respective plant extracts.

Ferulic acid (0.62), caffeic acid (0.5736), and p-coumaric acid (0.5116) were predicted to target the integrin α4β1 complex. Integrins are transmembrane receptors involved in cell adhesion and signal transduction. The α4β1 integrin has been implicated in the pathophysiology of epilepsy through its role in leukocyte adhesion and migration across the blood–brain barrier, contributing to neuroinflammation [41,42]. Modulation of this integrin could potentially attenuate inflammatory responses associated with seizure activity.

The target prediction analysis identifies several phytochemicals in the studied plant extracts that may interact with key molecular targets associated with epilepsy. Notably, compounds predicted to modulate GABAA and 5-HT3 receptors align with established mechanisms of anticonvulsant action, while interactions with integrin α4β1 suggest potential anti-inflammatory effects. Therefore, we chose the 5-HT3A receptor for further docking studies, considering its role as an emerging therapeutic target for the management of epileptic seizures.

#### 2.3.2. Molecular Docking

The docking studies focused on the 5-HT3A receptor, considering that GABA-A modulation by the selected plant extracts is already well-documented in the literature. In contrast, the potential interaction with 5-HT3A remains mainly unexplored, offering a novel mechanistic angle. Molecular docking simulations were conducted to explore the binding affinities and predicted interactions of selected phytochemicals with the mouse and human 5-HT3A receptors, both of which were co-crystallized with vortioxetine. The murine receptor structure (PDB ID: 8AW2) corresponds to the active conformation, while the human receptor (PDB ID: 8BLA) was used in its inactive state. The docking protocol was validated by redocking vortioxetine into each receptor’s binding site, resulting in RMSD values of 1.9174 Å for the mouse structure and 0.2254 Å for the human structure, confirming the reliability of the docking protocol (RMSD < 2.0 Å, Figure 4).

Among all screened phytochemicals, naringenin, apigenin, and pinocembrin demonstrated the strongest binding affinities across both species (Table 4). In the murine receptor, naringenin exhibited the lowest binding energy (−10.414 kcal/mol, LE = 0.5207), followed closely by apigenin (−10.277 kcal/mol, LE = 0.5139) and pinocembrin (−10.270 kcal/mol, LE = 0.5405). Similar binding energies were observed for the human receptor: naringenin (−10.223 kcal/mol), apigenin (−10.112 kcal/mol), and quercetin (−10.259 kcal/mol).

As shown in the docking analysis of naringenin, the compound engaged in multiple conventional hydrogen bonds within the mouse 5-HT3A binding pocket (Figure 5A,B). Specifically, hydrogen bonds were observed with Ala208 and Lys127, while π–π T-shaped and π-alkyl interactions involved residues Tyr126, Phe199, and Arg65. Moreover, a pi-donor hydrogen bond was formed with Asn101. In the human 5-HT3A receptor, naringenin formed hydrogen bonds with Arg191 and Tyr86, and they showed additional π–π stacking with Tyr148 and Ile66 (Figure 5C,D). The orientation and interaction pattern of the chromone scaffold resembled that of known 5-HT3 antagonists, potentially indicating functional antagonism.

Chlorogenic acid also demonstrated high binding affinity, especially in the mouse receptor (−9.758 kcal/mol). Docking visualization revealed hydrogen bonds with key polar residues, including Arg169, Trp156, and Asn101. A pi-donor hydrogen bond was also formed with Trp156, while the chlorogenic acid engaged in a pi-alkyl interaction with Ile201 (Figure 6A,B). In the human receptor, several residues contribute to hydrogen bonding: Thr176, Asn123, Trp178, Ser201, Asp199, and Arg191. (Figure 6C,D). A polar pi-cation interaction was also formed with Arg87, while a hydrophobic interaction was observed with Met223, reinforcing the potential of chlorogenic acid as a serotonergic modulator.

Overall, compounds such as naringenin, apigenin, chlorogenic acid, and pinocembrin exhibited high binding affinity and favorable interaction profiles with both mouse and human 5-HT3A receptors. These findings support their potential role in modulating serotonergic signaling, which has been implicated in the suppression of seizures and neuroprotection.

## 3. Discussion

The present study demonstrated the anticonvulsant potential of extracts from *Morus alba, Angelica archangelica*, *Passiflora incarnata*, and *Valeriana officinalis* through integrated phytochemical characterization, in vivo evaluations, biochemical assays, and computational predictions.

The selection of doses for each plant extract was based on previously published pharmacological studies evaluating their anticonvulsant activity in rodent models. For *Morus alba,* Gupta et al. (2014) reported effective anticonvulsant activity in pentylenetetrazol (PTZ)-induced seizure models in rats, using doses ranging from 25 to 100 mg/kg [19]. Accordingly, we selected 100 mg/kg as the upper safe and effective dose to maximize potential efficacy without reaching sedative or toxic thresholds. In the case of *Angelica archangelica,* Luszczki et al. (2007) demonstrated significant anticonvulsant effects of its primary active constituent, imperatorin, in the mouse maximal electroshock (MES) model at doses of 50–100 mg/kg [43]. Thus, a similar dose (100 mg/kg) was used in our study to reflect the active compound profile of the extract and align with the effective range reported in the literature. For *Passiflora incarnata*, previous studies showed anticonvulsant activity in strychnine-induced seizure models in mice at relatively low doses [44]. Since *Passiflora* extracts are known for their high GABA content and can exhibit CNS depressant activity, we selected a moderate dose of 50 mg/kg to minimize sedative interference with seizure thresholds while retaining efficacy. Finally, *Valeriana officinalis* has been studied in various rodent models, with effective doses ranging from 50 to 800 mg/kg, depending on the extract type and seizure model [33]. Since valepotriates, among its main active constituents, exhibit anticonvulsant effects at lower doses (5–20 mg/kg), we selected 50 mg/kg as a balanced dose expected to yield efficacy without strong sedative effects, which could confound the interpretation of seizure latency and severity. This rationale aimed to ensure that each plant extract was tested at a dose supported by the literature for its anticonvulsant potential while avoiding adverse behavioral suppression that could bias the results.

Among the tested extracts, *Morus alba* extract (MAE) exhibited the most pronounced anticonvulsant effects, significantly reducing both seizure duration (−92.86%, *p* = 0.0242) and incidence (50%, *p* = 0.0385). These findings align with previous studies reporting the anticonvulsant properties of Morus alba, attributed to flavonoids such as quercetin, kaempferol, and isorhamnetin, known for their antioxidant capabilities and modulatory actions on GABAergic neurotransmission [14,15,16]. Notably, MAE showed the highest content of these flavonoids, particularly isorhamnetin and quercetin, which might significantly contribute to its observed anticonvulsant activity. Isorhamnetin has been shown to promote neuronal differentiation by potentiating nerve growth factor-induced neurite outgrowth in PC12 cells [45] and to exert cerebrovascular protection in a murine model of ischemic stroke by reducing infarct volume and mitigating oxidative damage [46]. Quercetin, a flavonol with multiple hydroxyl groups, exhibits potent antioxidant effects by enhancing endogenous defense mechanisms, such as superoxide dismutase, glutathione peroxidase, and catalase activity, while also increasing glutathione levels and enhancing nitric oxide bioavailability [47,48]. These properties contribute to improved endothelial and neuronal function under oxidative stress. Moreover, quercetin has been shown to reduce seizure severity in kainic acid-induced convulsion models by downregulating the GABA-A receptor α5 subunit gene [49]. Chronic administration of quercetin (20–50 mg/kg, orally) has also conferred protection against seizures in models of alcohol withdrawal and PTZ-induced convulsions [50]. Taken together, the presence of these two compounds in MAE may synergistically enhance GABAergic modulation and antioxidant defense, thus contributing to its pronounced anticonvulsant efficacy in vivo.

*Angelica archangelica* (AAE) and *Passiflora incarnata* extracts (PIE) also exhibited moderate anticonvulsant effects, as indicated by reduced seizure duration. However, these results did not achieve statistical significance. The main constituents of AAE, such as ferulic acid, are recognized for their neuroprotective and anticonvulsant properties, which are mediated by anti-inflammatory pathways and the modulation of neuronal excitability [20,21,41,42]. Similarly, PIE’s anticonvulsant properties may stem from its flavonoid-rich composition, particularly apigenin, quercetin, and isorhamnetin, which have been previously reported to enhance inhibitory neurotransmission via GABA-A receptors [24,25,26,27]. However, the dosages employed might have been insufficient for statistically significant anticonvulsant effects, suggesting the need for further dose optimization.

*Valeriana officinalis* extract (VOE) exhibited minimal anticonvulsant efficacy, consistent with its lower concentration and diversity of phytochemical profile. Despite valerian’s historical use in managing neurological conditions due to its sedative and anticonvulsant properties [28,29,30,31,32,33], the specific preparation and dosing in the current study did not effectively demonstrate significant anticonvulsant activity.

Furthermore, for extracts that showed only moderate efficacy at the tested doses (AAE at 100 mg/kg and PIE at 50 mg/kg), we did not investigate higher dosage regimens that might reveal their full therapeutic potential. Therefore, it remains possible that a greater dose of PIE or AAE could yield significant anticonvulsant effects. However, dose escalation was constrained in this study to balance efficacy with safety.

Interestingly, despite variable anticonvulsant efficacy, all extracts significantly mitigated brain inflammatory responses and oxidative stress markers, as indicated by decreased TNF-α and increased total thiol levels. In particular, VOE exhibited the highest reduction in TNF-α (−39.64%) and substantial enhancement of thiol levels (+93.12%), suggesting strong anti-inflammatory and antioxidant activities. These biochemical effects could contribute to long-term neuroprotection, potentially benefiting chronic seizure management by reducing neuroinflammation and oxidative damage, both of which are critical factors in epileptogenesis [5,9,10,41].

The marked reduction in TNF-α levels observed with *V.officinalis* extract, despite its relatively modest anticonvulsant activity, can be attributed to its high chlorogenic acid content. Moreover, chlorogenic acids are a group of hydroxycinnamic acid derivatives, including caffeoylquinic, feruloylquinic, and dicaffeoylquinic acids, all of which are known for their potent anti-inflammatory and immunomodulatory properties [51,52]. These compounds have been shown to modulate key signaling pathways involved in inflammation, including NF-κB, JNK, ERK, and p38 MAPK [53,54]. Through these mechanisms, chlorogenic acid downregulates the expression of pro-inflammatory cytokines, such as TNF-α, IL-1β, IL-6, and IFN-γ, and reduces the levels of chemokines (e.g., MCP-1 and MIP-1α) [55,56]. Therefore, the anti-inflammatory effect of VOE is likely due to its rich chlorogenic acid profile, which may act independently of its central nervous system effects. Conversely, the weaker anticonvulsant response may stem from the inability of valerian’s primary active constituent, valerenic acid (a known GABA-A receptor modulator), to fully counteract the complex excitatory neurotransmitter release triggered by electroshock, including glutamate, acetylcholine, and adenosine [57]. These findings highlight a divergence between the anti-inflammatory and anticonvulsant efficacies of *Valeriana officinalis* and support its multifaceted pharmacological profile.

Target prediction and molecular docking analyses further supported the biological findings. Super-PRED predictions identified key molecular interactions with the GABA-A receptor (e.g., pinostrobin, formononetin, galangin, chrysin) and the 5-HT3 receptor (e.g., epicatechin gallate, pinocembrin). Both GABA-A and 5-HT3 receptors are recognized as significant anticonvulsant targets, modulating inhibitory neurotransmission and seizure susceptibility [38,39,40]. In our study, we prioritized the 5-HT3A receptor for docking, considering that GABA-A modulation by these plants is already well established in the literature. In contrast, the role of 5-HT3A in plant-based anticonvulsant activity remains largely unexplored, offering a novel opportunity for potential mechanistic exploration. Additionally, GABA-A receptors have multiple allosteric binding sites, making targeted docking less precise in the case of phytochemical binding. Focusing on 5-HT3A allowed for a more interpretable analysis and revealed strong predicted binding of key flavonoids.

Molecular docking highlighted the potential interactions of compounds such as naringenin, apigenin, chlorogenic acid, and pinocembrin with the mouse and human 5-HT3A receptors, showing favorable binding affinities and extensive interactions within critical receptor binding sites. These findings further support the therapeutic potential of these phytochemicals in managing seizures through serotonergic modulation, aligning with previous studies that emphasize the anticonvulsant relevance of 5-HT3 receptor interactions [39,40].

Moreover, several phytochemicals, including ferulic acid, caffeic acid, and p-coumaric acid, were predicted to modulate the integrin α4β1, which plays a role in neuroinflammation and blood–brain barrier integrity [41,42]. Such interactions suggest that these phytochemicals may provide additional neuroprotective benefits by attenuating inflammatory responses commonly associated with seizure disorders.

Recent studies have elucidated the modulatory effects of flavonoids on 5-HT_3_ receptors, which are ligand-gated ion channels implicated in various neurological processes, including seizure activity. Notably, quercetin has been shown to inhibit 5-HT-induced currents in 5-HT_3_A receptors expressed in *Xenopus* oocytes, with an IC_50_ of approximately 64.7 μM. This inhibition is characterized as competitive and voltage-independent, suggesting direct interaction with the receptor’s pre-transmembrane domain I [58]. Such findings highlight the potential of quercetin to modulate serotonergic neurotransmission, thereby contributing to its anticonvulsant properties. Similarly, apigenin has been shown to modulate serotonergic systems. In animal models, apigenin’s antidepressant-like effects were attenuated by the administration of ondansetron, a selective 5-HT_3_ receptor antagonist, indicating that its pharmacological actions may involve 5-HT_3_ receptor pathways [59]. These interactions suggest that apigenin may exert its neuroprotective effects, in part, by modulating 5-HT_3_ receptor activity. Collectively, these studies underscore the significance of flavonoid interactions with 5-HT_3_ receptors in the context of seizure modulation and neuroprotection. The integration of these findings with our current results reinforces the therapeutic potential of flavonoid-containing extracts in managing epilepsy and related neurological disorders.

One limitation of the present study is the lack of a dose-effect relationship analysis. Future studies are warranted to conduct dose-ranging experiments for each extract to determine the minimum effective dose and to investigate whether higher doses of AAE or PIE, for instance, could yield greater anticonvulsant effects in mice. Additionally, our study assessed anticonvulsant effects after only 7 days of extract administration, which is a relatively short duration. Since epilepsy is a chronic condition, it remains unknown whether long-term treatment with these extracts would sustain efficacy or reveal cumulative benefits, or even adverse effects. Therefore, chronic exposure studies are necessary to determine if the observed anti-seizure and neuroprotective effects persist or even improve over time. Such studies could also evaluate whether tolerance to the antiepileptic effects develops with prolonged use.

In conclusion, the study highlights *Morus alba* extract as the most promising candidate, exhibiting significant anticonvulsant activity that may be mediated by flavonoid interactions with GABA-A and 5-HT3 receptors, as well as notable anti-inflammatory and antioxidant effects. Although *Angelica archangelica* and *Passiflora incarnata* extracts showed moderate anticonvulsant effects, further dose-response studies could elucidate their full potential. *Valeriana officinalis* demonstrated strong anti-inflammatory and antioxidative properties but limited anticonvulsant efficacy under current experimental conditions. Future studies should investigate dose optimization, chronic administration effects, and detailed mechanistic pathways in both preclinical and clinical settings to fully assess the therapeutic viability of these extracts in epilepsy management.

## 4. Materials and Methods

### 4.1. Plant Material and Extracts

The roots of *Angelica archangelica* (used to prepare Angelica extract, AAE) and *Valeriana officinalis* (Valeriana extract, VOE), along with the aerial parts of *Passiflora incarnata* (Passiflora extract, PIE), were obtained in the form of herbal teas from commercial suppliers (Stef Mar Ltd., Râmnicu Vâlcea, and Fares, Orăștie, Romania, respectively). The bark of *Morus alba* (Morus extract, MAE) was collected in May 2018 from Buzău County, Romania. After initial air-drying at room temperature, the bark was further dehydrated in a conventional oven at 55 °C until it reached a brittle consistency suitable for grinding.

Each plant material was milled using a Swantech electric grinder, and the resulting powder was sieved through a 250 µm mesh screen. The extraction of *Morus alba* bark was performed using 70% ethanol (*v*/*v*). In contrast, the roots of *Angelica archangelica* and *Valeriana officinalis*, as well as the aerial parts of *Passiflora incarnata*, were extracted with 50% ethanol (*v*/*v*). All samples underwent double extraction under hot reflux conditions (approximately 75 °C). After filtration through standard filter paper, the pooled extracts were concentrated using a BÜCHI rotary evaporator at 70 °C. The concentrated residues were then lyophilized at −58 °C with a ScanVac CoolSafe freeze dryer (LaboGene, Lillerød, Denmark), yielding the final dry extracts.

### 4.2. Identification and Quantification of Phytochemicals by UHPLC-HRMS/MS

The qualitative and quantitative analysis of the extracts was conducted using ultra-high-performance liquid chromatography coupled with high-resolution tandem mass spectrometry (UHPLC-HRMS/MS). Analytical-grade standards, solvents (methanol and ethanol), formic acid, and ultrapure water were used in sample preparation. Stock solutions of individual reference compounds (1 mg/mL) were prepared in methanol and subsequently diluted to working concentrations ranging from 0.05 to 1 µg/mL. These solutions were stored at −20 °C before analysis.

Chromatographic separation was performed using a Dionex Ultimate 3000 UHPLC (Thermo Fisher Scientific, Waltham, MA, USA) system equipped with a binary pump, column oven, and autosampler. The stationary phase consisted of a C18 reversed-phase column (150 × 2.1 mm, 2.6 µm particle size). Mobile phase A consisted of water with 0.05% formic acid, while mobile phase B was methanol containing the same concentration of formic acid. A multistep gradient was applied at a flow rate of 0.3 mL/min and a column temperature of 40 °C.

Mass spectrometric detection was achieved using a Q-Exactive Orbitrap mass spectrometer (Thermo Fisher Scientific, Waltham, MA, USA) with a heated electrospray ionization (HESI) source. Source conditions included nitrogen gas flows for the sheath and auxiliary gases, a source temperature of 300 °C, a spray voltage of 2800 V, and an S-lens RF level set at 50. Full-scan spectra were recorded with a resolving power of 70,000 (FWHM at *m*/*z* 200), a mass range of *m*/*z* 100–1000, an AGC target of 3 × 10^6^, and an injection time of 200 ms. For structural elucidation, variable Data-Independent Acquisition (vDIA) was applied, with fragmentation across selected m/z windows using higher-energy collisional dissociation (HCD) at multiple collision energies (30, 60, and 80 NCE), followed by MS/MS detection at a resolution of 35,000.

Quantitative analysis was performed by comparing the results with calibration curves constructed from reference standards. When standards were unavailable, compound identification relied on high-accuracy mass measurements of precursor and product ions, matched with known fragmentation patterns. Molecular formulae were proposed based on minimal mass error (≤2 ppm), and structure elucidation was assisted by ChemSpider, NORMAN MassBank, mzCloud, and PubChem databases. For fragmentation pattern analysis, ACDLabs MS Fragmenter software (version 2019.2.1) was used [60,61].

### 4.3. In Vivo Studies

#### 4.3.1. Animals

All animal-related procedures adhered to the ethical principles established by Romanian Law No. 43/2014 concerning the protection of animals used for scientific purposes, as well as Directive 2010/63/EU of the European Parliament. The experimental design received ethical approval from the Bioethics Committee of the Faculty of Pharmacy, Carol Davila University of Medicine and Pharmacy, Bucharest, Romania (Approval No. 07/22 May 2020).

Forty-eight male NMRI mice (aged 6–8 weeks, body weight 25 ± 2.5 g) were sourced from the Cantacuzino National Institute of Research and Development for Microbiology and Immunology (INCDMI Cantacuzino), Bucharest. The animals were housed in standard plexiglass enclosures under controlled environmental conditions, with continuous access to drinking water and a rodent pellet diet provided by INCDMI Cantacuzino. The ambient temperature was maintained between 21 and 24 °C, with relative humidity ranging from 45% to 60%, as verified by hygrothermal monitoring. Before initiating the experimental procedures, all animals were allowed a 1-week acclimatization period in the facility.

#### 4.3.2. Treatment Groups

Animals were randomly divided into six experimental groups, each consisting of eight mice. Body weight was recorded on Day 0 and monitored every 2 days over the 8-day experimental period. Treatments were administered once daily by oral gavage.

The control group (CTL) received distilled water at a dose of 0.1 mL per 10 g of body weight (BW). The reference group was treated with sodium phenobarbital (PHB) at a dose of 25 mg/kg. The other four groups received plant-based dry extracts at the following concentrations: 100 mg/kg AAE, 100 mg/kg MAE, 50 mg/kg PIE, and 50 mg/kg VOE. All doses were selected based on previous literature supporting their pharmacological relevance and safety.

#### 4.3.3. Motor Activity

Mice were acclimated in the testing room for one hour before each session under controlled environmental conditions. Spontaneous motor activity was assessed on Day 0 (baseline), Day 7 (after six days of treatment), and Day 8 (after the final dose and recovery from tonic-clonic seizures).

Locomotor activity was recorded using a plexiglass chamber (40 × 40 × 25 cm) equipped with perpendicular infrared sensors (Activity Cage, Ugo Basile, Gemonio, Italy) to detect horizontal and vertical movements. Each beam break was automatically logged as a discrete movement.

Mice were tested individually for 5 min, with activity recorded at 1-minute intervals. Chambers were cleaned between sessions to eliminate olfactory interference. Total movement counts over 5 min were used to assess changes in motor activity due to treatment and seizure recovery [62].

#### 4.3.4. Anti-Seizure Activity

Anticonvulsant effects were evaluated on Day 8 using an electroshock-induced seizure model. Seizures were triggered with an ECT Unit (Ugo Basile, Gemonio, Italy) set to deliver a 15 mA current for 3 s at 100 pulses per second and 0.5 s pulse width—parameters optimized for mice weighing 25–30 g.

Before stimulation, auricular electrodes were soaked in Ringer’s solution to reduce contact resistance. Mice were gently restrained during electrode placement and then released before stimulation to prevent injury during seizures. The total duration of tonic-clonic seizures was recorded for each animal. Mean seizure durations, seizure incidence, and mortality were compared across treatment groups relative to controls to determine anticonvulsant efficacy [63].

#### 4.3.5. Biochemical Assays

Following cervical dislocation, brain tissues were collected from mice to assess TNF-α (inflammation marker) and total thiols (oxidative stress markers) concentrations. Brain samples were homogenized in a 1:10 weight-to-volume ratio using 0.25 M sucrose as the diluent. Homogenization was performed with an RW 14 basic homogenizer (IKA, Königswinter, Germany) to ensure consistent tissue disruption. Whole-brain homogenates were directly diluted 1:10 with phosphate-buffered saline (PBS) before biochemical assays. All samples were processed on ice to preserve enzymatic activity and prevent oxidative degradation.

The levels of TNF-α (catalog no. BMS 622) were evaluated by the instructions provided in the manual guide (Invitrogen, Thermo Fisher Scientific, Waltham, MA, USA). Total thiols were quantified using a previously described method, with a sample-to-Ellman reagent ratio of 1:4. Results are expressed as the amount of glutathione (GSH) equivalents (pM) per milligram of protein (mg/mL) [64].

### 4.4. Computational Studies

#### 4.4.1. Target Prediction

Molecular targets for the phytochemicals identified in the plant extracts were predicted using the Super-PRED web server (https://prediction.charite.de (accessed on 10 April 2025)). This in silico tool utilizes machine-learning algorithms and chemical structure similarity to predict potential protein targets based on existing bioactivity data. Among the predicted targets, those previously associated with seizure modulation or epilepsy-related pathways were selected for further investigation. These relevant targets were subjected to molecular docking studies to predict the binding affinities and potential molecular interactions for the identified compounds.

#### 4.4.2. Molecular Docking

Molecular docking studies were performed to investigate the potential interaction between phytochemicals identified by UHPLC-HRMS/MS and the 5-HT3A receptor, a target previously predicted using the Super-PRED web server. Two crystallographic structures were selected from the RCSB Protein Data Bank: the mouse 5-HT3A receptor in active conformation co-crystallized with vortioxetine (PDB ID: 8AW2) and the human 5-HT3A receptor in inactive conformation complexed with the same ligand (PDB ID: 8BLA) [65].

Protein structures were prepared using the YASARA Structure [66]. Water molecules and ligands were removed, structural errors were corrected, side chains were protonated at physiological pH (7.4), and hydrogen bonding networks were optimized. Energy minimization was conducted using the YASARA2 force field. Docking protocol validation was performed by redocking vortioxetine into its original binding site, and the accuracy of the docking was confirmed by calculating the root mean square deviation (RMSD) between the predicted and crystallographic poses. The identified phytochemicals were prepared by retrieving their SMILES codes from the PubChem database. Three-dimensional structures were generated using DataWarrior v5.2.1 [67], protonated at physiological pH, and energy-minimized with the MMFF94s+ force field.

Docking simulations were executed using AutoDock Vina v1.1.2 [68], targeting the vortioxetine-binding site in both receptor structures. Twelve docking runs were performed for each compound. Output parameters included predicted binding energy (ΔG, kcal/mol) and ligand efficiency (LE, ΔG per heavy atom). Ligand–receptor interactions were visualized and analyzed using BIOVIA Discovery Studio Visualizer (BIOVIA, Discovery Studio Visualizer, Version 17.2.0, Dassault Systèmes, 2016, San Diego, CA, USA).

### 4.5. Statistical Analysis

All statistical analyses were performed using GraphPad Prism version 6 (GraphPad Software Inc., San Diego, CA, USA). The normality of data distribution was assessed using the D’Agostino and Pearson omnibus test. For parametric datasets, differences between groups were analyzed using one-way ANOVA followed by Bonferroni’s post-hoc test. In cases where data were non-parametric, the Kruskal–Wallis test was applied, followed by Dunn’s multiple comparison test. Comparisons were made relative to the control group, and statistical significance was defined as a *p*-value of less than 0.05. To verify baseline homogeneity in locomotor activity before treatment, one-way ANOVA followed by Tukey’s post-hoc test was used. The incidence of seizures and mortality across groups was evaluated using Fisher’s exact test. For all experimental groups, results were expressed as mean ± standard error of the mean (SEM).

## 5. Conclusions

This study demonstrated that *Morus alba* extract exhibits significant anticonvulsant activity, likely mediated by its high flavonoid content and interactions with GABA-A and 5-HT3A receptors. Extracts of *Angelica archangelica* and *Passiflora incarnata* showed moderate, non-significant effects, while *Valeriana officinalis* displayed remarkable antioxidant and anti-inflammatory properties but limited seizure protection. Computational studies supported the involvement of serotonergic, GABAergic, and integrin α4β1 pathways. Overall, *Morus alba* shows the most significant potential as an adjuvant in epilepsy management, warranting further investigation in chronic seizure models and dose-optimization studies.

## Figures and Tables

**Figure 1 ijms-26-06426-f001:**
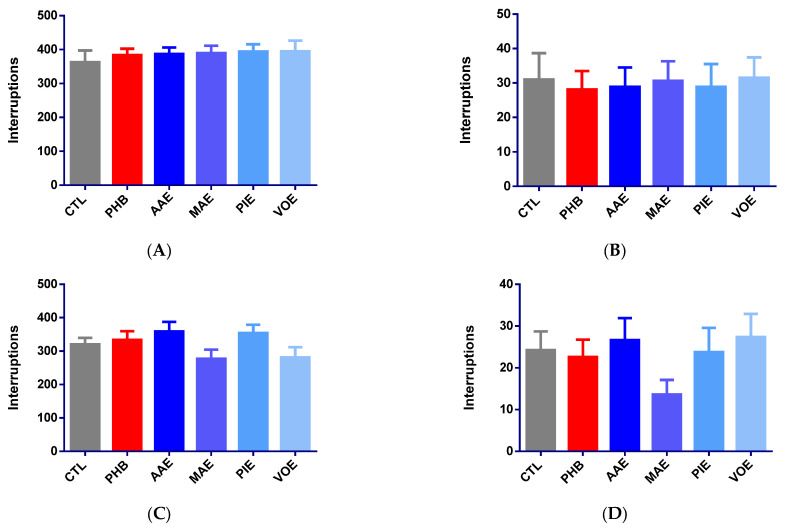
Variation of motor activity in mice, expressed as a mean number of beam interruptions ± SEM. (**A**) Baseline horizontal movements; (**B**) baseline vertical movements; (**C**) horizontal movements on Day 7; (**D**) vertical movements on Day 7; (**E**) post-seizure horizontal movements on Day 8; (**F**) post-seizure vertical movements on Day 9. * *p* < 0.05 vs. CTL; AAE—*A. archangelica* ethanolic extract; PIE—*P. incarnata* ethanolic extract; VOE—*V. officinalis* ethanolic extract; MAE—*M. alba* ethanolic extract; CTL—Control; PHB—Phenobarbital.

**Figure 2 ijms-26-06426-f002:**
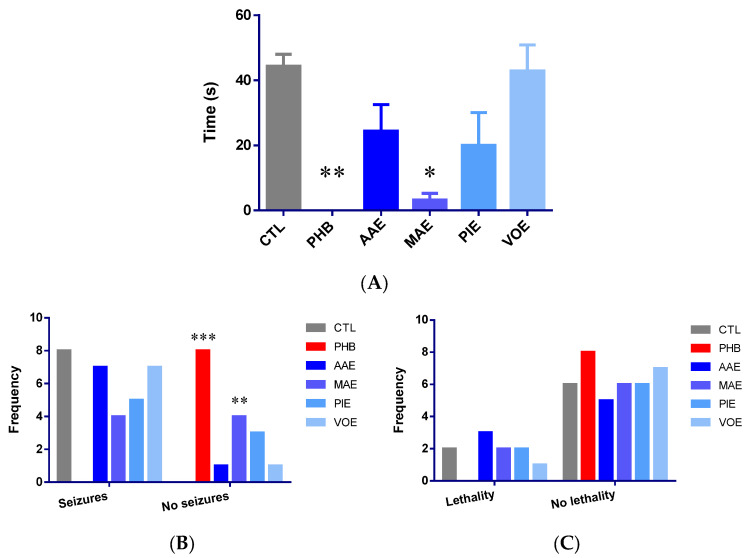
Anti-seizure activity in mice after 7 days of treatment. (**A**) Total tonic-clonic seizure duration (seconds) expressed as mean ± SEM; (**B**) variation of tonic-clonic seizure frequency; (**C**) variation of lethality frequency. * *p* < 0.05; ** *p* < 0.01, *** *p* < 0.001 vs. CTL; AAE—*A. archangelica* ethanolic extract; PIE—*P. incarnata* ethanolic extract; VOE—*V. officinalis* ethanolic extract; MAE—*M. alba* ethanolic extract; CTL—Control; PHB—Phenobarbital.

**Figure 3 ijms-26-06426-f003:**
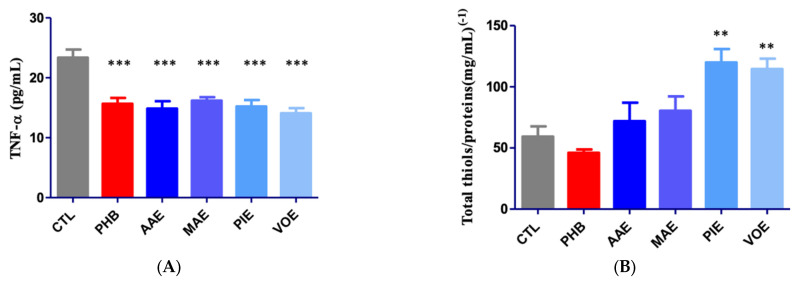
Quantification of inflammation and oxidative stress markers in mouse brain tissues. (**A**) Variation in TNF-α; (**B**) variation of total thiols/protein content. Values are expressed as mean + SEM ** *p* < 0.01, *** *p* < 0.001 vs. CTL; AAE—*A. archangelica* ethanolic extract; PIE—*P. incarnata* ethanolic extract; VOE—*V. officinalis* ethanolic extract; MAE—*M. alba* ethanolic extract; CTL—Control; PHB—Phenobarbital.

**Figure 4 ijms-26-06426-f004:**
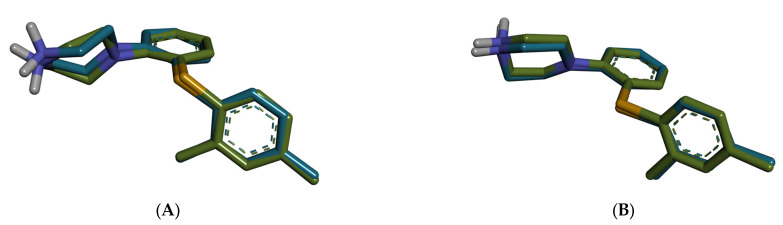
Superposition of the redocked pose (blue) of vortioxetine on experimental conformation (green). (**A**) mouse 5-HT3A (8AW2); (**B**) human 5-HT3A (8BLA).

**Figure 5 ijms-26-06426-f005:**
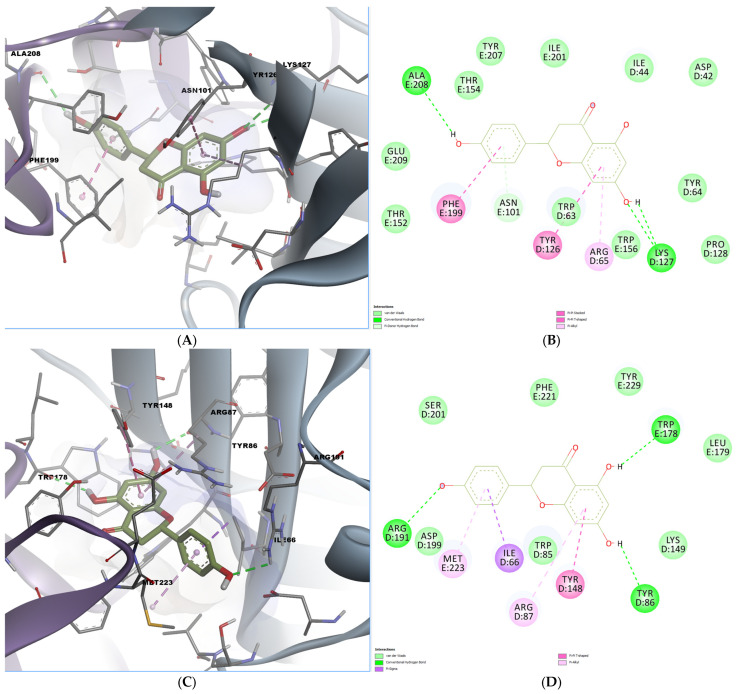
Predicted binding poses and interactions for naringenin in the 5-HT3A orthosteric binding site. (**A**) Docked pose in the mouse binding site; (**B**) 2D predicted interactions diagram between naringenin and the mouse binding site; (**C**) docked pose in the human binding site; (**D**) 2D predicted interactions diagram between naringenin and the human binding site.

**Figure 6 ijms-26-06426-f006:**
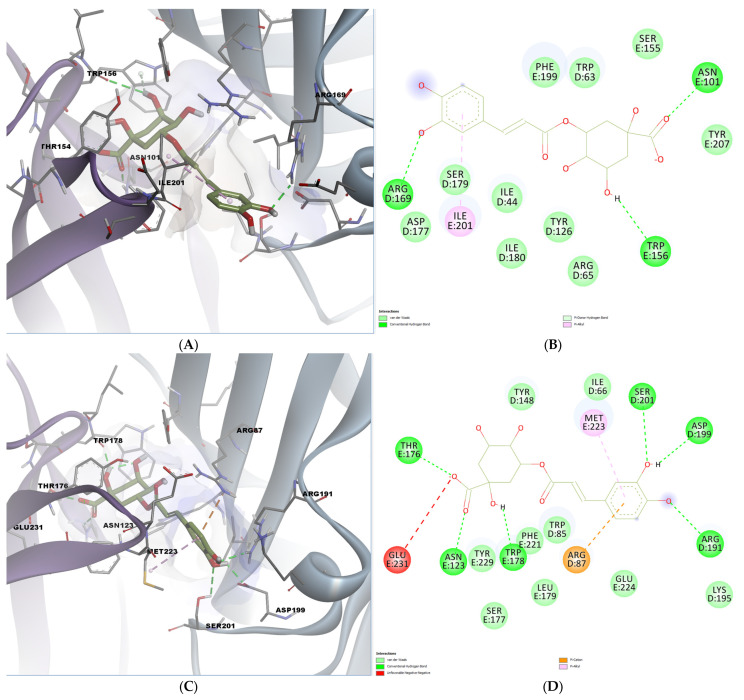
Predicted binding poses and interactions for chlorogenic acid in the 5-HT3A orthosteric binding site. (**A**) Docked pose in the mouse binding site; (**B**) 2D predicted interactions diagram between chlorogenic acid and the mouse binding site; (**C**) docked pose in the human binding site; (**D**) 2D predicted interactions diagram between chlorogenic acid and the human binding site.

**Table 1 ijms-26-06426-t001:** Phytocompounds identified in the plant extracts by UHPLC-HRMS/MS.

Compound	Chemical Formula	Exact Mass	Adduct Ion (*m*/*z*)	Retention Time (min)
Abscinic acid	C15H20O4	264.13616	263.12891	15.73
Apigenin	C15H10O5	270.05282	269.04502	17.54
Caffeic acid	C9H8O4	180.04226	179.03501	8.71
Catechin	C15H14O6	290.07904	289.07176	7.80
Chlorogenic acid	C16H18O9	354.09508	353.08783	8.20
Chrysin	C15H10O4	254.05791	253.05066	17.63
Daidzein	C15H10O4	254.05791	253.05066	16.50
Epicatechin gallate	C15H14O6	290.07904	289.07176	10.19
Ferulic acid	C10H10O4	194.05791	193.05066	14.98
Formononetin	C16H12O4	268.07356	267.06631	18.74
Galangin	C15H10O5	270.05282	269.04557	19.98
Gallic acid	C7H6O5	170.02152	169.01427	1.73
Genistein	C15H10O5	270.05282	269.04502	18.07
Genistin	C21H20O10	432.10565	431.09837	12.56
Glycitein	C16H12O5	284.06847	283.06122	16.33
Hesperitin	C16H14O6	302.07904	301.07179	16.83
Hyperoside	C21H20O12	464.09548	463.08768	17.63
Isorhamnetin	C16H12O7	316.0583	315.05105	13.20
Kaempferol	C15H10O6	286.04774	285.04049	17.06
Naringenin	C15H12O5	272.06847	271.06122	19.69
p-Coumaric acid	C9H8O3	164.04734	163.03954	10.77
Pinocembrin	C15H12O4	256.07356	255.06631	17.50
Pinostrobin	C16H14O4	270.08921	269.08196	17.40
Quercitin	C15H10O7	302.04265	301.0354	16.59
Rutin	C27H30O16	610.15338	609.14613	14.20
Syringic acid	C9H10O5	198.05282	197.04555	15.38

**Table 2 ijms-26-06426-t002:** Phytocompounds quantified in the plant extracts by UHPLC-HRMS/MS.

	Concentration (µg/g Extract)			
Compound	AAE	PIE	VOE	MAE
Abscinic acid	NF	152.41	NF	152.39
Apigenin	7.68	40.75	NF	164.87
Caffeic acid	3.43	NF	NF	NF
Catechin	NF	NF	NF	10,735.54
Chlorogenic acid	779.76	437.31	11,714.16	1761.08
Chrysin	NF	NF	NF	362.43
Daidzein	NF	20.50	NF	144.42
Epicatechin gallate	NF	NF	NF	NF
Ferulic acid	6350.87	402.99	NF	255.81
Formononetin	0.28	NF	NF	NF
Galangin	NF	NF	NF	311.67
Gallic acid	179.10	231.40	207.59	174.38
Genistein	70.9	NF	66.6	NF
Genistin	NF	93.45	NF	NF
Glycitein	NF	NF	NF	NF
Hesperetin	NF	NF	NF	NF
Hyperoside	NF	117.94	NF	2367.92
Isorhamentin	NF	1885.15	NF	61,936.48
Kaempherol	NF	102.14	NF	7935.80
Naringenin	11.45	9.89	NF	557.35
p-Coumaric acid	1223.13	3154.68	1098.99	2091.38
Pinocembrin	NF	NF	NF	8.66
Pinostrobin	66.67	NF	NF	NF
Quercitin	NF	5660.54	NF	26,356.80
Rutin	202.93	289.39	123.53	274.42
Syringic acid	NF	165.3	NF	NF

NF—not found; AAE—*A. archangelica* ethanolic extract; PIE—*P. incarnata* ethanolic extract; VOE—*V. officinalis* ethanolic extract; MAE—*M. alba* ethanolic extract.

**Table 3 ijms-26-06426-t003:** Predicted targets for the assessed phytochemicals.

Compound	Target Name	Probabilities
Abscisic acid	NA	NA
Apigenin	NA	NA
Caffeic acid	Integrin α4/β1	0.5736
Catechin	NA	NA
Chlorogenic acid	NA	NA
Chrysin	GABA-A receptor; α1/β2/γ2	0.7022
Daidzein	GABA-A receptor; α1/β2/γ2	0.5649
Epicatechin gallate	GABA-A receptor; α1/β2/γ2; Serotonin 3A (5-HT3A) receptor	0.6461; 0.6823
Ferulic acid	Integrin α4/β1	0.6200
Formononetin	GABA-A receptor; α1/β2/γ2	0.7516
Galangin	GABA-A receptor; α1/β2/γ2	0.7201
Gallic acid	NA	NA
Genistein	NA	NA
Genistin	NA	NA
Glycitein	GABA-A receptor; α1/β2/γ2	0.6125
Hesperetin	NA	NA
Hyperoside	NA	NA
Isorhamnetin	NA	NA
Kaempferol	NA	NA
Naringenin	NA	NA
p-Coumaric acid	Integrin α4/β1	0.5116
Pinocembrin	GABA-A receptor; α1/β2/γ2; Serotonin 3A (5-HT3A) receptor	0.6715; 0.5441
Pinostrobin	GABA-A receptor; α1/β2/γ2	0.7796
Quercetin	NA	NA
Rutin	NA	NA
Syringic acid	NA	NA

NA—not available.

**Table 4 ijms-26-06426-t004:** Molecular docking results for the screened phytochemicals.

	Mouse 5-HT3A		Human 5-HT3A	
Compound	Binding Energy (kcal/mol)	LE	Binding Energy (kcal/mol)	LE
Abscisic acid	−7.886	0.4151	−9.739	0.5126
Apigenin	−10.277	0.5139	−10.112	0.5056
Caffeic acid	−8.231	0.6332	−8.496	0.6535
Catechin	−8.756	0.417	−10.195	0.4855
Chlorogenic acid	−9.758	0.3903	−9.882	0.3953
Chrysin	−8.784	0.4623	−9.864	0.5192
Daidzein	−10.109	0.5321	−9.276	0.4882
Epicatechin gallate	−8.042	0.2513	−6.699	0.2093
Ferulic acid	−7.72	0.5514	−8.075	0.5768
Formononetin	−8.991	0.4281	−8.909	0.4242
Galangin	−8.918	0.4459	−9.897	0.4949
Gallic acid	−6.606	0.5505	−6.859	0.5716
Genistein	−9.126	0.4563	−9.209	0.4604
Genistin	−9.517	0.307	−9.538	0.3077
Glycitein	−9.041	0.4305	−9.142	0.4353
Hesperetin	−8.533	0.4063	−9.39	0.4471
Hyperoside	−7.422	0.2249	−7.482	0.2267
Isorhamnetin	−9.191	0.3996	−10.205	0.4437
Kaempferol	−8.712	0.4149	−9.962	0.4744
Naringenin	−10.414	0.5207	−10.223	0.5111
p-Coumaric acid	−8.017	0.6681	−8.006	0.6672
Pinocembrin	−10.27	0.5405	−9.859	0.5189
Pinostrobin	−8.866	0.4433	−9.746	0.4873
Quercetin	−9.076	0.4125	−10.259	0.4663
Rutin	−7.564	0.1759	−7.023	0.1633
Syringic acid	−6.758	0.4827	−6.609	0.4721

## Data Availability

The data presented in this study are available on request from the first corresponding author and the last author.

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
