# Peer review of "Investigation of Anticonvulsant Potential of Morus alba, Angelica archangelica, Valeriana officinalis, and Passiflora incarnata Extracts: In Vivo and In Silico Studies"

_ijms, 2025, doi:10.3390/ijms26136426_

Round 1

Reviewer 1 Report

Comments and Suggestions for Authors

Comment 1: Add the dosages used for each extract in the abstract, which is critical for interpreting the varying anticonvulsant effects.

Comment 2: The introduction lacks a detailed review of recent literature, especially from the last 2-3 years, which could contextualize the study’s novelty.

Comment 3:  The introduction lacks the recent literature on plant based biomedical applications. I recommend the inclusion of a recent study in the introduction section:

J.N. Rana, S. Mumtaz, Prunin: An Emerging Anticancer Flavonoid, Int. J. Mol. Sci. 26 (2025). https://doi.org/10.3390/ijms26062678.

Comment 4. Why were Morus alba, Angelica archangelica, Passiflora incarnata, and Valeriana officinalis chosen over other anticonvulsant herbs? Discuss this in introduction section by describing their key features with objectives of this research.

Comment 5. The study uses different doses (100 mg/kg for AAE and MAE, 50 mg/kg for PIE and VOE) without justification. Provide a clear justification for this.

Comment 6. In Figure 1, provide significance for each group to understand the degree of effects. If the effect is not significant, mention as “ns”.

Comment 7: Explain, how MAE’s high isorhamnetin and quercetin content correlates with its superior anticonvulsant effects.

Comment 8. Discuss why VOE, with minimal anticonvulsant activity, showed the highest TNF-α reduction.

Comment 9. Justify prioritizing 5-HT3A over GABA-A for docking and align findings with in vivo results.

Comment 10. Provide units along y-axis of graphs in Figure 1 and 2. If the unit is arbitrary, mention as “a.u.”.

Comment 11: Minor grammatical errors and inconsistent phrasing exist. Conduct thorough proofreading to enhance clarity in the revised version.

Author Response

Response for Reviewer 1 Report

We are grateful for your time and attention, as well as for your accurate and valuable comments, which aim to improve the quality of the present manuscript. We responded point by point to each comment and would be pleased to know that we succeeded. 

We thank Reviewer 1 for their thorough evaluation of our manuscript. We have carefully addressed each of the comments as detailed below.

Comment 1: Add the dosages used for each extract in the abstract, which is critical for interpreting the varying anticonvulsant effects.

R1: We agree that including the extract dosages in the Abstract will improve clarity. We have modified the Abstract to specify the dose administered for each plant extract, so that readers immediately understand the treatment conditions.

Comment 2: The introduction lacks a detailed review of recent literature, especially from the last 2-3 years, which could contextualize the study’s novelty.

R2: We appreciate this suggestion. We have expanded the Introduction to incorporate recent literature, thereby better highlighting the novelty and context of our study. For instance, we cited a 2025 study regarding the phenolic profile and antioxidant activity of Morus alba infusions (lines 82-84) and other recent literature throughout the manuscript.

Comment 3: The introduction lacks the recent literature on plant-based biomedical applications. I recommend the inclusion of a recent study in the introduction section:

J.N. Rana, S. Mumtaz, Prunin: An Emerging Anticancer Flavonoid, Int. J. Mol. Sci. 26 (2025). https://doi.org/10.3390/ijms26062678.

R3: We thank Reviewer 1 for the valuable comment. We incorporated the suggested reference, and we added a paragraph in the Introduction section to highlight recent literature on plant-based biomedical applications (lines 55-71)

Comment 4.  Why were Morus alba, Angelica archangelica, Passiflora incarnata, and Valeriana officinalis chosen over other anticonvulsant herbs? Discuss this in introduction section by describing their key features with objectives of this research.

R4: We realize that our original Introduction described each plant’s properties, but did not explicitly state why these four were selected. We have now added a clarifying explanation of our rationale for choosing these species. Please see lines 123-145 in the Introduction section.

Comment 5. The study uses different doses (100 mg/kg for AAE and MAE, 50 mg/kg for PIE and VOE) without justification. Provide a clear justification for this.

R5. We agree that the manuscript needed a more explicit justification for the dosing of each extract. We have expanded the Discussion section to justify each selected dose, while citing supporting literature. Please see Discussion section, lines 557-579.

Comment 6. In Figure 1, provide significance for each group to understand the degree of effects. If the effect is not significant, mention as “ns”.

R6: We thank Reviewer 1 for this specific observation. The only significant effect was observed for the PHB group compared to the control, regarding post-seizure horizontal movements (Figure 1E). This significant difference is marked with an asterisk (*) (p < 0.05) on the figure panel, with an explanation provided in the figure caption. However, we chose not to mark as “ns” all the other results, as it would add too much clutter to the figures; the absence of any symbol or explanation is generally considered a non-significant result.

Comment 7: Explain, how MAE’s high isorhamnetin and quercetin content correlates with its superior anticonvulsant effects.

R7: We thank Reviewer 1 for this suggestion. We have expanded our Discussion to more explicitly connect the phytochemical profile of Morus alba extract with its strong anticonvulsant efficacy, providing a more detailed explanation. Please see lines 587-603 in the Discussion section.

Comment 8. Discuss why VOE, with minimal anticonvulsant activity, showed the highest TNF-α reduction.

R8: We thank Reviewer 1 for this valuable comment, and we agree that this point deserves further discussion. In the revised manuscript, we address the seemingly paradoxical result for Valeriana officinalis extract (i.e., strong anti-inflammatory effect due to TNF-α reduction) despite weak seizure protection. Please see lines 632-661 in the Discussions section.

Comment 9. Justify prioritizing 5-HT3A over GABA-A for docking and align findings with in vivo results.

R9: Thank you for pointing this shortcoming out, since we indeed did not clarify this aspect in the original version. We now explain the rationale for prioritizing the 5-HT3A receptor over GABA-A in the Results and Discussion sections. Please see lines 487-490 and 670-677.

Comment 10. Provide units along y-axis of graphs in Figure 1 and 2. If the unit is arbitrary, mention as “a.u.”.

R10: We thank Reviewer 1 for this observation. However, in Figure 1, ‘interruptions” on y-axis, as described in the figure caption, refers to the number of beam interruptions or breaks in locomotor activity assays. This measure represents the actual number of events or event counts; therefore, it should not be labeled as arbitrary units. Moreover, Figures 2B and 2C illustrate the number of subjects, labeled as frequency on the y-axis. If the figures had instead illustrated the percentage of subjects, then the unit should have been (%).

Comment 11: Minor grammatical errors and inconsistent phrasing exist. Conduct thorough proofreading to enhance clarity in the revised version.

R11: We thank Reviewer 1 for bringing these issues to our attention. The manuscript has been thoroughly proofread and edited for grammar, clarity, and consistency.

Reviewer 2 Report

Comments and Suggestions for Authors

The authors of the submitted manuscript undertook a multifaceted study of the anticonvulsant activity of four plant extracts, using phytochemical methods, in vivo studies and in silico analyses. The aim of the work is clearly defined and justified by the clinical problem of drug resistance in epilepsy. The methodology is comprehensive and well-planned. The results are consistent and correctly presented. The statistical analysis is carried out correctly. The interpretation of the results is accurate and supported by data. Despite the good substantive level of the submitted work, I have several comments:
- the article does not present a justification for the doses of plant extracts used, nor does it analyze the dose-effect relationship
- in the case of extracts with moderate efficacy (AAE, PIE), the authors did not test higher doses, which makes it difficult to assess their full therapeutic potential.
- lack of data on chronic exposure - assessment of the effect only after 7 days is insufficient in the context of epilepsy, which is a chronic disease.
- lack of information on the standardization of extracts (e.g. content of the main flavonoids relative to the mass of the extract).

Author Response

Response for Reviewer 2 Report

We are grateful for your time and attention, as well as for your accurate and valuable comments, which aim to improve the quality of the present manuscript. We responded point by point to each comment and would be pleased to know that we succeeded. 

The authors of the submitted manuscript undertook a multifaceted study of the anticonvulsant activity of four plant extracts, using phytochemical methods, in vivo studies and in silico analyses. The aim of the work is clearly defined and justified by the clinical problem of drug resistance in epilepsy. The methodology is comprehensive and well-planned. The results are consistent and correctly presented. The statistical analysis is carried out correctly. The interpretation of the results is accurate and supported by data. Despite the good substantive level of the submitted work, I have several comments:

We thank Reviewer 2 for their thorough evaluation of our manuscript and for the positive assessment of our work. We have carefully addressed each of the comments as detailed below.

Comment 1. The article does not present a justification for the doses of plant extracts used, nor does it analyze the dose-effect relationship.

R1: We appreciate this important observation. We have now added a clear justification for the extract doses in the Discussion section (please see lines 560-582). We also acknowledge in the Discussion that our study used a single fixed dose for each extract and did not include a dose–response analysis. This is indeed a limitation, and we have now explicitly noted that future studies should evaluate multiple doses to fully characterize the dose-effect relationship (please see lines 707-710).

Comment 2. In the case of extracts with moderate efficacy (AAE, PIE), the authors did not test higher doses, which makes it difficult to assess their full therapeutic potential.

R2: We thank Reviewer 2 for this valuable insight. To address this issue, we expanded the Discussion section to acknowledge that higher doses of AAE and PIE were not explored, which leaves some uncertainty regarding their maximum therapeutic potential. Please see lines 622-626.

Comment 3. Lack of data on chronic exposure - assessment of the effect only after 7 days is insufficient in the context of epilepsy, which is a chronic disease.

R3: We thank Reviewer 2 for this observation. We acknowledge that our experiment evaluated subacute treatment (daily dosing for 7 days and then seizure induction on day 8), which models acute seizure protection and short-term biochemical changes. Epilepsy, however, is a chronic condition, and the long-term efficacy and safety of these extracts were not determined in our study. In response, we have added a clear statement in the Discussion section about this limitation. Please see lines 710-730.

Comment 4. Lack of information on the standardization of extracts (e.g. content of the main flavonoids relative to the mass of the extract).

R4: We apologize if the extract composition was not emphasized clearly in the initial manuscript version. To increase the readability of quantification results, we converted high phytochemical concentrations from micrograms per gram of extract to milligrams per gram (mg/g). Moreover, for compounds detected in large amounts in the extracts (e.g., isorhamnetin in MAE), the percentage related to the exact extract mass was also reported. Please see lines 204-227.

Round 2

Reviewer 1 Report

Comments and Suggestions for Authors

The authors have addressed all of my comments and concerns in the revised version. I recommend accepting the paper in present form.

Reviewer 2 Report

Comments and Suggestions for Authors

I recommend the revised manuscript for publication and IJMS.